# Ticagrelor vs Clopidogrel in addition to Aspirin in minor ischemic stroke/ transient ischemic attack—Protocol for a systematic review and network meta-analysis

**Gabriele Zitikyte**[1,2‡], **Danielle Carole Roy**[1‡], **Shan Dhaliwal**[1☯], **Ronda Lun**[1,3☯]*, **Brian Hutton**[1,2], **Risa Shorr**[4], **Dar Dowlatshahi**[1,2,3]

**1** University of Ottawa, School of Epidemiology and Public Health, Ottawa, Ontario, Canada, **2** Ottawa Hospital Research Institute, Clinical Epidemiology Program, Ottawa, Ontario, Canada, **3** Division of Neurology, Department of Medicine, The Ottawa Hospital, Ottawa, Ontario, Canada, **4** Department of Education, The Ottawa Hospital, Ottawa, Ontario, Canada

☯ These authors contributed equally to this work.
‡ These authors contributed equally to this manuscript as co-first authors
* rlun@toh.ca

**Funding:** The authors received no specific funding for this work.

**Competing interests:** The authors have declared that no competing interests exist.

## Abstract

### Introduction

Patients with minor ischemic stroke or transient ischemic attack represent a high-risk population for recurrent stroke. No direct comparison exists comparing dual antiplatelet therapy regimens—namely, Ticagrelor and Aspirin versus Clopidogrel and Aspirin. This systematic review and network meta-analysis (NMA) will examine the efficacy of these two different antiplatelet regimens in preventing recurrent stroke and mortality up to 30 days.

### Methods and analysis

MEDLINE, EMBASE, and the Cochrane Central Register of Controlled Trials (CENTRAL) will be searched with the assistance of a medical information specialist. Two independent reviewers will screen studies for inclusion; eligible studies will include randomized controlled trials that enrolled adults presenting with acute minor ischemic stroke or transient ischemic attack and compared one or more of the interventions against each other and/or a control. The primary outcomes will be recurrent ischemic stroke up to 30 days from symptom onset. Secondary outcomes will include safety outcomes (I.e. major bleeding and mortality), functional disability, and outcomes up to 90 days from symptom onset. A Bayesian approach to NMA will be implemented using the BUGSnet function in R Software. Between group comparisons for time-to-event (TTE) and dichotomous outcomes will be presented in terms of hazard ratios and odds ratios with 95% credible intervals, respectively. Secondary effect measures of treatment ranking will also be estimated.

### Ethics and dissemination

No formal research ethics approval are necessary. We will disseminate our findings through scientific conference presentations, peer-reviewed publications, and social media/the

press. The findings from this review will aid clinicians in decision-making on the choice of antithrombotic therapy in a high-risk stroke population and could be important in the development of future treatment trials and guidelines.

Registration ID with Open Science Framework: 10.17605/OSF.IO/XDJYZ.

## Introduction

Patients with minor ischemic stroke or transient ischemic attack (TIA) represent a population at high risk for recurrent stroke [1–3]. The recurrent stroke rate in the next 3 months following an initial event has been suggested to be anywhere between 10–20% [4]. Clinical worsening with a subsequent event is common, and may result from progression of the initial infarct, or recurrent ischemic events [4]. Identifying optimal secondary prevention strategies in this high-risk population is crucial as it may reduce morbidity and mortality [5].

Recent clinical trials involving high-risk stroke subjects such as the POINT and CHANCE trials have established the superior efficacy of dual antiplatelet therapy (DAPT) with Clopidogrel and Aspirin compared to single antiplatelet therapy with Aspirin alone [6,7]. These trials have necessitated changes to multiple stroke treatment guidelines, recommending treatment with dual antiplatelet therapy for patients presenting with high-risk minor stroke/TIA [8,9]. Clopidogrel is a prodrug that is activated in two steps to irreversibly inhibit the binding of ADP to the $P2Y_{12}$-receptor [10]. In the literature, significant variability in response to Clopidogrel exists, largely due to the prevalence of genetic polymorphisms (i.e.CYP2C19 loss of function alleles *2, *3, and *8) that render the metabolism of Clopidogrel ineffectively [11]. The prevalence of such genetic polymorphisms is reported to be especially high in Asian populations—up to 60% [12]. Clinically, this has been associated with an increased risk for "treatment failure" leading to recurrent ischemic events [12].

An alternative antiplatelet medication is Ticagrelor—an antiplatelet agent that does not require metabolic activation, but rather directly binds and inhibits platelet $P2Y_{12}$ receptors [13,14]. A recent trial evaluated the use of Ticagrelor and Aspirin in the treatment of the same high-risk patient population [14]. It was found that the combination of Ticagrelor and Aspirin was superior to Aspirin monotherapy in reducing the risk for stroke or death—the event rate was 6.6% in the DAPT group compared to 5.5% in the Aspirin only group. However, the efficacy of DAPT with Ticagrelor and Aspirin has never been directly compared to an alternative regimen of Clopidogrel and Aspirin.

While traditional pairwise meta-analysis is of great value and familiarity, network meta-analysis (NMA) is a vital methodology that is able to cohesively analyze multiple comparators of relevance based on indirect evidence which does not exist in the primary literature. This protocol describes the methodology for a systematic review and network meta-analysis that will assess the relative effects of competing treatments for patients with acute minor ischemic stroke or TIA in terms of prevention of recurrent ischemic events, death, safety profiles (including major hemorrhage rates [15] and mortality), and the proportion of patients achieving functional independence.

## Materials & methods

### Study question, registration, and reporting

The review will address the following research question: What is the efficacy of ticagrelor and aspirin in preventing recurrent ischemic strokes compared to clopidogrel and aspirin?

The review protocol has been submitted to the Open Science Framework for registration (https://osf.io/xdjyz). Reporting of the completed review will be prepared in consultation of the PRISMA Extension Statement for Network Meta-Analysis. Any protocol deviations incurred will be described in the final study report.

## Search strategy

The search strategy for this review will utilize the following databases: Medline (OVID interface), EMBASE (OVID interface), and Cochrane (OVID interface). The search will include articles from database inception until the February 2021. The search strategy, developed with the assistance of a health science librarian with expertise in systematic reviews, can be found in the S2 File. Additionally, we will search the abstracts database from the World Stroke Congress and International Stroke Conference in the last 20 years for potentially relevant abstracts. If such is found, we will contact the authors of the abstract to inquire about obtaining data.

## Inclusion and exclusion criteria

Detailed inclusion and exclusion criteria are listed in Table 1.

## Data management

All study records will be stored in a Sharepoint folder shared amongst the review team for simultaneous access to data and study files. This will include study protocol, documentation of screening, eligibility, search terms as well as completed data extraction forms and risk of bias assessments. Covidence Systematic Review Software (Covidence, Melbourne, VIC, Australia) will be used for screening and data extraction, as well as deletion of duplicate studies.

## Selection process

Data will be collected by two independent reviewers for each phase of the review including screening, eligibility and extraction. Study selection will be done in two phases—the first will comprise of title and abstract screening only, and the second will comprise of full-text screening. If a study meets all of the inclusion criteria and none of the exclusion criteria, it will be included for extraction. At each phase, reviewers will evaluate the completeness, content and quality of the studies. If data is missing/unclear for a study, reviewers will contact the study investigators to obtain further information. The data collection will be done using a standardized electronic data collection form (S1 File) shared amongst the review team. Trial authors may be contacted if there are inadequate details allowing clear judgement of bias in a domain.

## Data extraction

Data extraction will be undertaken by both content experts and methodologists. For each study/article, there will be two reviewers independently extracting the appropriate data. Any disagreements regarding the extracted data will be resolved via discussion between the reviewers, with consultation of a third party if necessary, to resolve the discrepancy. Where encountered, multiple reports for a single study will be extracted separately during the data extraction process, but will be collated and linked together for the analysis.

**Data items.**   The data items extracted will be from the following categories:

- Study Characteristics: Title, First author, Publication year, Journal, Country of origin, Funding

**Table 1. Inclusion/exclusion criteria for the planned systematic review.**

| Inclusion Criteria | Population:<br>• Adult patients only (18 years or older)<br>• Presenting stroke severity of National Institute of Health Stroke Severity Scale (NIHSS) of 5 or less or a TIA ABCD2 score of 4 or higher<br>• Must have started treatment within 72 hours of presenting stroke or TIA<br><br>Intervention:<br>• Ticagrelor in combination with Aspirin in any dose or formulation<br><br>Comparator:<br>• Clopidogrel in combination with Aspirin in any dose or formulation<br><br>Outcomes:<br>• Reports recurrent ischemic event or death as an outcome with a minimum length of follow-up of at least 30 days post-stroke<br><br>Study Factors:<br>• Randomized controlled trials only<br>• Language: English or French |
|---|---|
| Exclusion Criteria | Population:<br>• Pediatric populations<br>• Patients receiving thrombolysis or endovascular thrombectomy<br>• Non-acute stroke (I.e. > 24 hours from symptom onset to randomization)<br><br>Intervention:<br>• Use of antiplatelet agents other than Clopidogrel, Ticagrelor, or Aspirin<br><br>Comparator:<br>• Use of antiplatelet agents other than Clopidogrel, Ticagrelor, or Aspirin<br><br>Outcomes:<br>• Diagnosis of other subtype of stroke, including all intracerebral hemorrhage or cerebral venous sinus thrombosis<br><br>Study Factors:<br>• Non-RCT study/article: cohort studies, cross-sectional studies, survey studies, reviews, etc.<br>• Grey literature studies<br>• Language: not in English or French<br>• Overlapping trial populations will be dealt with by only including the study with the largest N |

- <u>Study Methodology</u>: Study Design, Study Setting, Allocation Sequence Concealment, Inclusion/Exclusion

- <u>Participant Characteristics</u>: sample size, age, sex, country, % of previous stroke, follow-up period and % of TIA or minor ischemic stroke

- <u>Intervention/Comparator</u>: number of participants in each group, drug, route of administration, drug dose, drug loading dose, drug maintenance dose, treatment frequency, duration

- <u>Outcomes</u>: recurrent stroke (yes or no), mortality (yes or no), bleeding events (yes or no), modalities used for diagnosis, definitions

  - Ischemic stroke definition: neurologic dysfunction caused by focal cerebral infarction confirmed by neuroimaging or pathology [16]

- Bleeding definition: bleeding events and method used to ascertain the bleeding event will be recorded for each study. If definitions vary between studies, we will assess heterogeneity across studies as needed.

- Results: number of participants included in analysis, number of participants lost to follow-up, % of outcome(s) in intervention/comparator groups, number of adverse events, at two separate time points (30 days and 90 days), person-time at risk for each outcome

The final data extraction dataset will be reviewed by two other reviewers to ensure the data was entered correctly.

## Risk of bias assessment

In this systematic review, risk of bias of individual studies will be assessed using the Cochrane Risk of Bias tool for randomized trials (RoB2) [17]. If there are any crossover or cluster designed randomized trials, we will use the respective RoB variant for these trials (RoB2 for crossover trials and RoB2 for cluster-randomized trials). To implement the risk of bias assessment, we will use the RoB2 excel tool which is to be done by two independent raters. If there is disagreement in the rating that cannot be resolved by discussion, the final decision will be made by consulting a third team member. The RoB2 assessment will consider the effect of assignment (also known as the intention to treat effect) as the effect of interest in which will target the results of the primary outcome: recurrent stroke. The main domains of the RoB2 assessment tool include: random sequence generation, effect of assignment, missing outcome data, measurement of the outcome and selective outcome reporting. These domains will be judged using "high, some concerns or low" risk of bias measures. The overall risk of bias will be determined by the measures of the domains. If all domains are judged low risk, the overall measure of bias will be low for this study. If there are some concerns in at least one domain with no high risk judged domains, the overall measure of bias will remain as "some concerns". If a domain is judged as high risk or has many domains judged to have some concerns, this study will receive an overall score of high risk of bias. Raters will also be reviewing the studies for other possible biases such as sponsorship or publication bias. Trial authors may be contacted if there are inadequate details allowing clear judgement of bias in a domain.

## Statistical analysis

**Descriptive analysis.** Population characteristics and methodological homogeneity for all eligible RCTs will be reviewed and summarized descriptively by the research team. If the trials are judged to be sufficiently homogenous, a pairwise meta-analysis per treatment comparison will be performed to evaluate homogeneity. We assume that patients who are included in studies that are eligible for this review are equally likely to be randomized to either of the dual antiplatelet therapies that we plan to compare.

**Primary outcome.** The primary outcome of interest will be recurrent ischemic stroke (Time-To-Event) up to 30 days. 30 days was chosen as the primary outcome because the majority of recurrent ischemic stroke events in this population occur within the first 21 days; data from trials investigating DAPT in this population have found that DAPT does not significantly reduce stroke risk during days 22–90 [18]. Time-To-Event (TTE) data will be presented as hazard ratios within intention-to-treat populations with 95% confidence intervals. If hazard ratios are not available for a study, they will be estimated using hazard ratio extrapolation methods presented by Guyot et al [19]. The proportion of patients with the primary outcome at 30 days will be compared between treatments using a chi-square test with logistic regression modeling to control for covariates.

**Secondary outcomes.** Secondary outcomes will include safety outcomes such as major bleeding, mortality (both TTE) and functional disability (dichotomous). We will additionally look at recurrent ischemic stroke events and bleeding events up to 90 days as secondary outcomes. TTE and dichotomous endpoints will be expressed as hazard ratios and odds ratios, respectively, with 95% confidence intervals.

**Pairwise meta-analysis.** A standard meta-analysis will be performed for each pairwise comparison of antiplatelet combinations. Hazard ratios for time-to-event outcomes and odds ratios for dichotomous outcomes, with 95% confidence intervals, will be obtained and between-study heterogeneity will be quantified using the $I^2$ measure.

## Methods for network meta-analyses

**The transitivity assumption.** A key underlying assumption of NMA is transitivity (otherwise referred to as homogeneity and similarity), such that competing interventions are jointly randomizable and effect modifiers do not differ between them [20,21]. In collaboration with our clinical experts, age, stroke severity, baseline mRS, and history of hypertension and diabetes have been identified as important effect modifiers that will be appraised for homogeneity by the study team. The distribution of these variables across studies will be investigated, and we will further explore whether other patient characteristics are balanced across trials. We will assess the impact of covariates through subgroup analyses and/or meta-regression adjustments, chosen in collaboration with clinical experts in the field. These may include but are not limited to: stroke severity and onset to randomization according to inclusion criteria of the initial trial, country of publication, and year of publication.

Transitivity will be attenuated by only including studies for which methodology and characteristics are as similar as possible [22]. This decision will be informed by careful consideration of study methods (eg. follow-up, etc) and patient characteristics (eg. eligibility and demographics such as stroke severity, age, baseline mRS, etc) by the research team. We will also perform pairwise meta-analyses for each direct comparison of interventions using the available studies to quantify statistical heterogeneity using the I2 measures to assess further heterogeneity between studies. If heterogeneity is judged to be excessive, findings will be presented via a narrative summary with supporting tables and figures. If homogeneity is sufficient, both fixed- and random-effects NMAs will be performed to compare interventions contained within the included studies and account for any effect modification [23].

**Consistency.** Consistency (also 'Coherence') is the statistical manifestation of transitivity and assesses whether the direct and indirect effect estimates from closed loops in the network are in agreement [20,24,25]. The relative effects on an appropriate scale must "add-up"; for example, the log-hazard ratio for the comparison of antiplatelet therapies A vs C is the sum of the log-hazard ratio for therapies A vs B and B vs C. Consistency will be evaluated by fitting an unrelated means model and comparing the model fit statistics and residuals with the consistency model.

**Network meta-analysis.** Network meta-analyses (NMA) allow synthesis of results from trials of interventions that form a connected network so that direct and indirect evidence can be statistically combined [21]. All analyses will be performed within Bayesian framework using BUGSnet 1.0.4, and evaluated with Markov Chain Monte Carlo simulation [23,24,26]. We will compare residual deviance with the number of unconstrained data points to assess model fit and, if quantities are approximately equal, the model fit will be deemed adequate. The selection between models will be based on deviance information criteria (DIC), with smaller values indicative of a greater fit and a difference greater than five points suggesting an important difference [27,28]. Convergence will be assessed using Gelman and Rubin criteria and by

inspecting trace plots [26,28]. NMA also enables ranking of treatments according to the probability that each is the best, second best, and so on, for a given outcome. Probabilistic statistics will be calculated for each intervention and results will be plotted [29].

## Ethics and dissemination

No formal research ethics approval will be necessary for this study as primary data will not be collected. The findings from this review and network meta-analysis will aid clinicians in decision-making on the choice of antithrombotic therapy in a high-risk stroke population and could be important in the development of future treatment trials and guidelines. Therefore, we will disseminate our findings through a combination of scientific conference presentations, peer-reviewed publications, and social media/the press.

## Conclusions

This review is planned as there is evolving evidence suggesting variable effectiveness of antiplatelet therapies for patients with minor ischemic stroke or TIA. Traditional systematic reviews and meta-analyses exist that compare Ticagrelor or Clopidogrel against Aspirin alone; however, no network meta-analyses enabling the comparison of both indirect and direct evidence have been performed [6,7,14]. This review will be the first to compare both DAPT therapies head-to-head and provide relative effectiveness, thus facilitating evidence-based management of patients suffering from a minor ischemic stroke or TIA, and identifying key areas for future research. We will prioritize patient-important outcomes such as recurrent strokes, mortality, and bleeding events.

## Supporting information

**S1 Checklist.**
(DOCX)

**S1 File.**
(XLSX)

**S2 File.**
(DOCX)

## Acknowledgments

We would like to express our sincere gratitude to our professors, Dr. Dean Fergusson, MHA, PhD, FCAHS, Full Professor, Departments of Medicine & Surgery, & School of Epidemiology and Public Health, University of Ottawa, and Dr. Matthew McInnes, MD, FRCPC, BSc, PhD, Professor, Departments of Radiology, & School of Epidemiology and Public Health, University of Ottawa, for teaching us about systematic reviews and providing guidance throughout this protocol development.

## Author Contributions

**Conceptualization:** Gabriele Zitikyte, Danielle Carole Roy, Shan Dhaliwal, Ronda Lun.

**Investigation:** Shan Dhaliwal, Ronda Lun.

**Methodology:** Gabriele Zitikyte, Danielle Carole Roy, Ronda Lun, Brian Hutton, Risa Shorr.

**Supervision:** Dar Dowlatshahi.

**Writing – original draft:** Gabriele Zitikyte, Danielle Carole Roy, Shan Dhaliwal, Ronda Lun.

**Writing – review & editing:** Gabriele Zitikyte, Danielle Carole Roy, Shan Dhaliwal, Ronda Lun, Dar Dowlatshahi.

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
