## [Decision Letter · Decision Letter 0]

1 Apr 2021

PONE-D-21-08104

Ticagrelor vs Clopidogrel in addition to Aspirin in minor ischemic stroke/ transient ischemic attack – protocol for a systematic review + network meta-analysis

PLOS ONE

Dear Dr. Lun,

Thank you for submitting your manuscript to PLOS ONE. After careful consideration, we feel that it has merit but does not fully meet PLOS ONE’s publication criteria as it currently stands. Therefore, we invite you to submit a revised version of the manuscript that addresses the points raised during the review process.

We look forward to receiving your revised manuscript.

Kind regards,

Aristeidis H. Katsanos, MD, PhD

Academic Editor

PLOS ONE

Journal Requirements:

"The funders had and will not have a role in study design, data collection and analysis,

decision to publish, or preparation of the manuscript."

4. We note you have included a table to which you do not refer in the text of your manuscript. Please ensure that you refer to Table 1 in your text; if accepted, production will need this reference to link the reader to the Table.

Reviewers' comments:

Reviewer's Responses to Questions

**Comments to the Author**

1. Does the manuscript provide a valid rationale for the proposed study, with clearly identified and justified research questions?

Reviewer #1: Yes

Reviewer #2: Yes

2. Is the protocol technically sound and planned in a manner that will lead to a meaningful outcome and allow testing the stated hypotheses?

Reviewer #1: Yes

Reviewer #2: Partly

3. Is the methodology feasible and described in sufficient detail to allow the work to be replicable?

Reviewer #1: Yes

Reviewer #2: Yes

4. Have the authors described where all data underlying the findings will be made available when the study is complete?

Reviewer #1: No

Reviewer #2: No

5. Is the manuscript presented in an intelligible fashion and written in standard English?

Reviewer #1: Yes

Reviewer #2: Yes

6. Review Comments to the Author

You may also provide optional suggestions and comments to authors that they might find helpful in planning their study.

Reviewer #1: The authors present an interesting protocol aimed at comparing dual antiplatelet treatment in stroke recurrence prevention after minor stroke or TIA.

The protocol is sound, methods are well defined and eligibility criteria similar to previous reviews.

Few points to address:

- Regarding grey literature, please define the reasons underlying the exclusion of other repositories (Rxiv and similars, or search engine as GScholar).

- Regarding medications chosen, please detail the a-priori exclusion of other antithrombotics (e.g. cilostazol)

Please also consider to provide the tabular view of data extracted through accessible repositories, as well as the syntax script used for analysis to allow reproducibility of the research.

Reviewer #2: Thank you for the opportunity to review this protocol for meta-analysis and review. The study provides the planned protocol for a network meta-analysis of different DAPT regiments post-TIA, specifically Ticagrelor + Aspirin vs Clopidogrel + Aspirin. The authors provide good reasoning for the necessity of this type of analysis, and provide the required information for the study protocol.

The authors suggest going through abstracts from ISC and WSO. It would also be a good idea to go through the abstracts of the European Stroke Organisation Conferences.

While it is fully up to the authors to decide the data extraction complexity, the fact that this study does not use individual patient level data is a study design drawback that limits the conclusions of this study. As most RCT's have generous data-sharing policies, it should not be too difficult to receive these datasets given the very reasonable scientific question of this study. A comment on the limitation of not having individual patient level data should be added.

While the authors write in the “Transitivity assumption” how heterogeneity will be handled, there is no specification on what criteria for heterogeneity will be used. This section should be expanded.

THALES and POINT have differing inclusion/exclusion criteria in terms of stroke severity and TIA risk score. This will be an inherent difference between studies which will still be within the inclusion criteria of this meta-analysis. It would be of interest to perform a subgroup analysis of POINT-like patients within the THALES population, if the extracted data will allow it.

Additional between study effect modifiers that should be considered are: Onset to randomization, onset to treatment, sex.

Assessing balance of potential effect modifiers is a good start but will still be lacking because you do not have individual patient level data. You will not be able to assess if patients with multiple effect modifiers, for example hypertension + diabetes, have a synergistic effect on the outcomes. While these effects should be balanced through randomization, there is no guarantee that they are balanced between studies. A brief comment on this is needed, perhaps within the previous IPD comment.

The GRADE framework level of evidence is not included as a inclusion/exclusion criteria. Hence, the sentences describing that only RCT's of high quality will be included yet that they can also be downgraded in evidence grade are confusing. This part should be clarified or removed.

7. PLOS authors have the option to publish the peer review history of their article (what does this mean?). If published, this will include your full peer review and any attached files.

Reviewer #1: No

Reviewer #2: No

---

## [Author Response · Author response to Decision Letter 0]

8 Apr 2021

Response to Reviewers

Reviewer #1: The authors present an interesting protocol aimed at comparing dual antiplatelet treatment in stroke recurrence prevention after minor stroke or TIA.

The protocol is sound, methods are well defined and eligibility criteria similar to previous reviews.

Few points to address:

- Regarding grey literature, please define the reasons underlying the exclusion of other repositories (Rxiv and similars, or search engine as GScholar).

Thank you for your comments. While we appreciate the breadth of literature in gray literature repositories, the objective of this systematic review is to include only evidence of the highest standard, and therefore we chose to only include peer-reviewed journals and publications of the randomized controlled trial type. We recognize this is a limitation of the current study. 

We are open to adding a “Limitations” paragraph to our protocol paper, but would prefer to include it in the final publication instead, since we anticipate this will ultimately be a more comprehensive reflection of the limitations of the study. However, we have drafted up a limitations paragraph (see page 4) and will leave the decision to the reviewers/ editor in terms of its inclusion in the current protocol. 

- Regarding medications chosen, please detail the a-priori exclusion of other antithrombotics (e.g. cilostazol)

We chose to exclude all other antithrombotics, including anticoagulants (i.e. warfarin, direct oral anticoagulants), and all other formulations of antiplatelet medications that are not Clopidogrel, Ticagrelor, or Aspirin. 

While other antiplatelet medications have been tested in the minor stroke/ TIA population (including cilostazol), the only recommended dual antiplatelet therapy in this population currently from the 2018 American Heart Association guidelines for secondary prevention after minor ischemic stroke/ TIA are for Clopidogrel and Aspirin. However, with the latest evidence from THALES (comparing Ticagrelor and aspirin), there may be emerging evidence for use of Ticagrelor – in fact, we have confirmed that the revised 2021 Canadian Stroke Best Practice recommendations (in press) will include Ticagrelor in their recommendations. We expect the American guidelines (and others) will likely change soon to account for the new emerging evidence for Ticagrelor. Therefore, we chose specifically to compare Ticagrelor and Aspirin vs Clopidogrel and Aspirin. However, we recognize there are alternative antiplatelet regimens recommended by other guidelines, and therefore we will include this as a limitation in our final manuscript.

Please also consider to provide the tabular view of data extracted through accessible repositories, as well as the syntax script used for analysis to allow reproducibility of the research.

We will make all source documents available upon request, in accordance to PLoS ONE publication policies. Thank you for your suggestion.

Reviewer #2: Thank you for the opportunity to review this protocol for meta-analysis and review. The study provides the planned protocol for a network meta-analysis of different DAPT regiments post-TIA, specifically Ticagrelor + Aspirin vs Clopidogrel + Aspirin. The authors provide good reasoning for the necessity of this type of analysis, and provide the required information for the study protocol.

The authors suggest going through abstracts from ISC and WSO. It would also be a good idea to go through the abstracts of the European Stroke Organisation Conferences.

Thank you for this great suggestion. We have added ESOC to the list of abstracts we review. 

While it is fully up to the authors to decide the data extraction complexity, the fact that this study does not use individual patient level data is a study design drawback that limits the conclusions of this study. As most RCT's have generous data-sharing policies, it should not be too difficult to receive these datasets given the very reasonable scientific question of this study. A comment on the limitation of not having individual patient level data should be added.

We agree that individual patient data (IPD) meta-analyses would be ideal. However, we are concerned that the process of retrieving and collating individual patient data would significantly delay the completion of this systematic review. For example, the POINT trial has specifically stated in their study protocol (section 12): “Three years after the primary publication associated with this work is submitted, a HIPAA-compliant, de-identified version of the database will be made available publicly.” Since this trial was published in 2018 this would mean that we would have to wait until at least November 2021 to gain access to this data. Moreover, in our prior experience with IPDs, some studies will require research ethics board amendments to share their data, which can again add to completion time. As this is an emerging and important topic currently, we would prefer to perform an aggregate level meta-analysis first, and subsequently explore an IPDMA. 

While the authors write in the “Transitivity assumption” how heterogeneity will be handled, there is no specification on what criteria for heterogeneity will be used. This section should be expanded.

Thank you for the suggestion. Our protocol has been updated accordingly. 

See lines 235 – 240 of the revised manuscript: 

“This decision will be informed by careful consideration of study methods (eg., follow-up, etc) and patient characteristics (eg., eligibility and demographics such as stroke severity, age, baseline mRS, etc) by the research team. We will also perform pairwise meta-analyses for each direct comparison of interventions using the available studies to quantify statistical heterogeneity using the I2 measures to assess further heterogeneity between studies.”

THALES and POINT have differing inclusion/exclusion criteria in terms of stroke severity and TIA risk score. This will be an inherent difference between studies which will still be within the inclusion criteria of this meta-analysis. It would be of interest to perform a subgroup analysis of POINT-like patients within the THALES population, if the extracted data will allow it.

Additional between study effect modifiers that should be considered are: Onset to randomization, onset to treatment, sex.

Unfortunately, because we are not performing an IPD-level meta-analysis, we are unable to perform specific subgroup analysis based on initial sex. However, we plan on performing subgroup analyses based on the median NIHSS/ TIA risk score and time to randomization reported on a study level. Furthermore, we will be stratifying patients by stroke severity and onset to randomization as per trial inclusion criteria, year of publication, and country of publication, and have updated our protocol to reflect these changes. 

Assessing balance of potential effect modifiers is a good start but will still be lacking because you do not have individual patient level data. You will not be able to assess if patients with multiple effect modifiers, for example hypertension + diabetes, have a synergistic effect on the outcomes. While these effects should be balanced through randomization, there is no guarantee that they are balanced between studies. A brief comment on this is needed, perhaps within the previous IPD comment.

We recognize this is a limitation of our aggregate-level meta-analysis. We have expanded our limitations paragraph to account for this. 

The GRADE framework level of evidence is not included as a inclusion/exclusion criteria. Hence, the sentences describing that only RCT's of high quality will be included yet that they can also be downgraded in evidence grade are confusing. This part should be clarified or removed.

This section has been removed as per the reviewer’s suggestions.

 

Limitations Paragraph – to be included at the reviewers/ editor’s discretion 

There are a number of limitations that must be considered when examining this systematic review. First, by focusing on randomized trials, we may overlook potential data from non-randomized studies and gray literature. However, we believe this is also a strength of our study, as it focuses only on article types with the highest hierarchy of evidence (i.e. RCT). Second, we did not utilize individual patient data for this study which limits the number of subgroup analyses that we are able to perform. This is unfortunately a limitation that we cannot avoid in order to conduct a timely review, as some clinical trials (e.g. POINT) have not yet released their trial data at the time of developing this study protocol. Third, given that the inclusion/exclusion criteria differ slightly between studies, there may be residual clinical and statistical heterogeneity between studies, which will be assessed by performing and reporting I2 statistics in the final manuscript. Lastly, it is also possible that reporting or publication bias may be present in certain studies, which will be assessed and reported using the Risk of Bias 2 assessment tool.

---

## [Editor Report · Decision Letter 1]

12 Apr 2021

Ticagrelor vs Clopidogrel in addition to Aspirin in minor ischemic stroke/ transient ischemic attack – protocol for a systematic review + network meta-analysis

PONE-D-21-08104R1

Dear Dr. Lun,

We’re pleased to inform you that your manuscript has been judged scientifically suitable for publication and will be formally accepted for publication once it meets all outstanding technical requirements.

Kind regards,

Aristeidis H. Katsanos, MD, PhD

Academic Editor

PLOS ONE
---

## [Editor Report · Acceptance letter]

14 Apr 2021

PONE-D-21-08104R1 

Ticagrelor vs Clopidogrel in addition to Aspirin in minor ischemic stroke/ transient ischemic attack – protocol for a systematic review and network meta-analysis 

Dear Dr. Lun:

I'm pleased to inform you that your manuscript has been deemed suitable for publication in PLOS ONE. Congratulations! Your manuscript is now with our production department. 

Kind regards, 

on behalf of

Dr. Aristeidis H. Katsanos 

Academic Editor

PLOS ONE